# Characteristics of Spatiotemporal Variations and Driving Factors of Land Use for Rural Tourism in Areas That Eliminated Poverty

**Yuanli Liu [1,2], Heping Liao [2], Jiqing Qiu [1,*] and Yan Liu [3]**

[1] School of Business Administration, Chongqing Technology and Business University, Chongqing 400067, China; lyl2021@ctbu.edu.cn

[2] Center for Targeted Poverty Alleviation and Regional Development Assessment, Southwest University, Chongqing 400715, China; liaohp@swu.edu.cn

[3] School of Public Affairs, Chongqing University, Chongqing 400044, China; lyspeaking@126.com

[*] Correspondence: lyjiqin@ctbu.edu.cn; Tel.: +86-133-4077-3797

**Abstract:** This study explores the spatiotemporal characteristics and driving factors of land use for rural tourism in areas that eliminated poverty from 2009 to 2021. It puts forward targeted governance measures to promote the high-quality development of rural tourism, poverty alleviation, and rural revitalization. The analysis is based on exploratory spatial analysis methods and geographical detectors. The results show that (1) the overall level of land use for rural tourism was low but grew very quickly with large regional differences. (2) There was a significant spatial agglomeration in land use for rural tourism land. The spatial distribution of land use for rural tourism landscapes was characterized by two cores and four clusters, while spatial distribution of rural tourism facilities was characterized by one cluster with multiple branches. (3) The driving factors of spatial variations in land use for rural tourism were diverse and dynamic. Dominant factors shifted from natural conditions and geographical location to socioeconomic and tourism resources and regional policy dimensions. Policy should emphasize the development of the rural tourism industry, innovate the diversified "tourism + development" model, enhance the level of land use for rural tourism, broaden avenues for farmers to increase their income, and strengthen residents' motivation for development.

**Keywords:** land use for rural tourism; rural industry revitalization; spatial distribution characteristics; areas that eliminated poverty

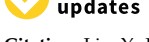



## 1. Introduction

On 16 October 2022, General Secretary Xi Jinping highlighted in his report at the 20th National Congress of the Chinese Communist Party that industries with unique rural characteristics should be developed to broaden avenues for farmers to increase their income and gain affluence. This would help to consolidate and expand the success of poverty alleviation and strengthen the intrinsic motivation for development in areas where poverty was eliminated and people were lifted out of poverty. Rural tourism is an industry with unique rural characteristics, characterized by a distinctive mode of production based on lifestyle and rural scenery [1]. It is an important means of consolidating and expanding the success of poverty alleviation in the new era and effectively links with the comprehensive promotion of rural revitalization [2]. Land resources form the foundation of rural tourism development, and optimal spatial allocation of tourism infrastructure will influence its development goals and social benefits. However, various problems, such as spatial differences in the utilization of land resources at a village level and planning delays, have led to low land use for rural tourism and poor spatial allocation, which hinders the development of rural tourism. Therefore, it is of great practical importance to consolidate and expand the success of poverty alleviation and bridge it with rural revitalization strategies. This can be

achieved by studying the characteristics of spatiotemporal variations, their driving factors, and changes in land use for rural tourism in areas that eliminated poverty to promote the development of rural tourism.

In the 1930s, McMurry was among the first scholars to examine the correlation between tourism development and land use and concluded that tourism constitutes a unique form of land use [3]. Scholars have since discussed the concept of land use for rural tourism from various perspectives, such as "point", "line", and "surface" [4], arguing that land use for tourism is an important medium in tourism development as it is legally used by rural tourism companies [5]. This improves the multi-functionality of land use [6], in terms of both commercial land for rural tourism and non-commercial land such as agricultural land and roads [7]. Academics have mainly conducted research on the efficiency of land use for tourism, the characteristics of its spatial distribution, and influencing factors. For example, Xian et al. analyzed the competitiveness of land use for rural tourism in Miyun District, Beijing, using a neural network method and weighted model to identify obstacles [8]. Overseas scholars have used spatial data to explore the impact of land-use patterns and intensity on the landscape [9,10] and sustainable management methods [11–13]. While academics have yet to reach a consensus regarding the classification of land use for rural tourism, the basic direction is relatively consistent. For example, Ning divided land use for rural tourism into built-up land and non-built-up land [14], while Lobo categorized it into land use for the tourism landscape and land use for tourism service facilities [15]. Yu et al. divided land use for tourism into different systems: resources, users, base, and interconnected land [16]. Weaver divided the different types of land in winter resorts into ski stations, ski areas, cable cars, and ring-shaped mountain walls [17]. Individual scholars have also explored the impact of land transfer on the development of rural tourism [18–20], examined the relationship between land use and rural tourism [21], investigated the relationship and conflict between red tourism resources and the spatial planning of land [22], and emphasized the promotion of sustainable land use for tourism through reasonable planning [23,24] and the implementation of rural revitalization strategies [25].

This study reviews and summarizes the concepts behind land use for rural tourism. Local culture and tourism activities are fundamental resource carriers of the rural natural landscape and a branch of land-use function involving land for rural tourism service facilities and landscape facilities. The research results of land use for rural tourism show that a preliminary framework and methodology have been developed, albeit without consensus on its concept, and its micro-level perspective and long-term longitudinal research still need to be strengthened. From a geospatial approach, few studies have analyzed the characteristics of spatiotemporal variations in land use for rural tourism in areas that eliminated poverty or explored the driving factors and their dynamic changes.

Therefore, using Rangtang County, Ngawa Tibetan, and Qiang Autonomous Prefecture, Sichuan Province as the case study for this research, 58 administrative villages were studied using exploratory spatial analysis methods and geographical detectors. The characteristics of spatiotemporal variations, driving factors, and changes in land use for rural tourism between 2009 and 2021 were investigated while actively exploring tourism as a means of poverty alleviation [26–28] to promote the consolidation and expansion of poverty alleviation success in these regions and linking it to China's rural revitalization strategy.

## 2. Materials and Methods

### 2.1. Overview of Study Area

Rangtang County in the Ngawa Tibetan and Qiang Autonomous Prefecture, Sichuan Province, is located on the southeastern edge of the Qinghai–Tibetan Plateau and on the upper reaches of the Dadu River. The county is dominated by plateaus, with an average altitude of 3285 m. Its annual average temperature is only 4.70 °C. It has cool summers, with a maximum temperature of 29.40 °C, and cold winters, with a minimum temperature of −23.40 °C. The poor transportation conditions in the study area, with a total of more than 900 km of roads, hinder the development of rural tourism. Due to strict controls by

the Chinese government, between 2009 and 2021, the area of cultivated land and fields increased significantly to 3791.63 and 2.57 km$^2$, respectively, while forest land increased slightly to 309,400 km$^2$, and grassland decreased slightly to 298,400 km$^2$. Furthermore, the area covered by urban villages, land for industries and mining, and other activities has been significantly reduced. Rangtang's tourism resources comprise a combination of natural and cultural landscapes, with natural resources, including the Nanmoqie Wetland and Xiangla Dongji Holy Mountain, and cultural resources, including Risimanba Diaofang and Juenang Cultural Center. There are eight key protected cultural relic units across all levels. Rangtang County, which was considered a poverty-stricken county at a national scale, successfully lifted itself out of poverty in 2020 and was identified as a key county for national rural revitalization in 2021. Hence, Rangtang County is a representative case study for studying areas that eliminated poverty.

### *2.2. Data Sources and Data Processing*

This study studied 58 administrative villages in Rangtang County in 2009, 2015, and 2021. First, the field data were completed in December 2021 through the Participatory Rural Assessment (PRA). The overall reliability of the scale was 0.791, the Kaiser–Meyer–Olkin value of the farmers' data reached 0.771 (greater than 0.6), and the Bartlett's sphericity was significant ($p < 0.05$), indicating that the reliability test and validity test of the research questionnaire met the requirements of scientific research (Devellis, 1991). Second, vector data were used, which included land-use data (2009, 2015, and 2021) and a digital elevation model of a 10 × 10 m precision derived from the County Natural Resources Bureau. Third, statistical data were collected; sources included the Rangtang County Statistical Yearbook (2009–2021) and the 2009–2021 economic statements of rural townships from the County Statistics Bureau. All the data were subjected to dimensionality reduction by principal component analysis (PCA) to ensure a scientific and objective development of the index system [29]. At the same time, the original data of the indicators were processed with range standardization, possible weight sets were constructed based on game theory, and the weight values of the indicators and dimensions were determined.

### *2.3. Research Methods*

#### 2.3.1. Model for Measuring the Level of Land Use for Rural Tourism

Based on the concept of land use for rural tourism, this study calculated the level of land use for the rural tourism landscape using four indicators—fields, forest land, grassland, and water bodies—and calculated the level of land use for rural tourism facilities using five indicators—rural roads, farmhouses, homestays, village squares, and scenic spots. Next, the level of rural tourism land use (RTL) was determined as follows:

$$RTL = \sum_{i=1}^{x} \left( \sum_{j=1}^{y} T_{ij} W_{ij} \right) W_i \tag{1}$$

In Equation (1), $x$ is the number of dimensions; $y$ is the number of indicators in the corresponding dimension; $T_{ij}$ is the standardized index value; $W_{ij}$ is the index weight; and $W_i$ is the dimension weight.

#### 2.3.2. Exploratory Spatial Data Analysis Method

The exploratory spatial analysis method constructed the spatial relationship between the research unit of the village and its surrounding neighborhoods by establishing spatial

weights and uncovered the characteristics of spatial correlation among attributes of the village research unit [30]. The calculation is as follows:

$$\text{Global Moran's I} = \frac{\sum\limits_{i}^{n}\sum\limits_{j\neq i}^{n} W_{ij}(X_i - X)(X_j - X)}{S^2 \sum\limits_{i}^{n}\sum\limits_{j\neq i}^{n} W_{ij}} \tag{2}$$

$$\text{Local Moran's I} = \frac{(X_i - X)}{S^2} \sum_{j=1}^{n} W_{ij}(X_j - X) \tag{3}$$

In Equation (2), $n$ is the 58 administrative villages in the research area; $X_i$ and $X_j$ are the utilization levels of research units $i$ and $j$, respectively; $X$ is the average value of the research units; and $W_{ij}$ is the distance between spatial units $i$ and $j$ in the spatial weight matrix.

2.3.3. Geographical Detectors

With the aid of the geographical detector model [31], the index of determination $q$ was introduced to detect the driving factors of land use for rural tourism. The $q$ value of the index of determination of each factor on land use for rural tourism land is:

$$q = 1 - \frac{1}{n\sigma^2} \sum_{h=1}^{L} n_h \sigma_h^2 \tag{4}$$

In Equation (3), $n_h$ is the number of samples in factor $h$ (corresponding to one or more sub-regions); $n$ is the number of research units; $\sigma^2$ is the discrete variance of the research area; and $L$ is the value of factor classification. The larger the value of the determining power $q$, the stronger the factor's degree of influence.

2.3.4. Model of Dynamic Changes in Influencing Factors

Drawing on existing research [32], the degree of dynamic change in the driving factors of land use for rural tourism was used to reflect the evolving state of these driving factors in the study area from 2009 to 2021. This provided a basis for a dynamic analysis of their impact mechanisms and the formulation of measures and is calculated as follows:

$$F = \frac{P_{i+1} - P_i}{P_i} \times 100\% \tag{5}$$

In Equation (4), $F$ is the degree of dynamic change (%) in the driving factors of land use for rural tourism; and $P_{i+1}$ and $P_i$ represent the determining power of a driving factor of land use for rural tourism in the $i$ + 1th year and $i$ year, respectively.

**3. Results**

*3.1. Characteristics of Spatiotemporal Variations in Land Use for Rural Tourism*

3.1.1. Overall Level of Land Use for Rural Tourism

The research results indicate that the overall level of land use for rural tourism in the study area was low, with large regional differences but nevertheless an increasing trend (Table 1). The average levels of land use for rural tourism in 2009 and 2015 were 0.29 and 0.35, respectively, and increased to 0.42 in 2021, which was 44.80% and 18.36% higher than in 2009 and 2015, respectively. This was mainly due to the implementation of poverty alleviation and rural revitalization strategies in the research area and improvements in rural infrastructure. The area covered by rural roads in the region increased by 210.41% between 2021 and 2009. From the perspective of the two dimensions of land use, the level of land use for rural tourism facilities grew rapidly, increasing from 0.31 to 0.39 and 0.47 in 2009, 2015, and 2021, respectively. The growth rates in 2015 and 2019 were 20.74% and 25.56%, respectively. The level of land use for rural tourism facilities in 2021 was 1.52 times that of

2009. However, the level of land use for the rural tourism landscape increased slightly, at 0.27, 0.32, and 0.37 in 2009, 2015, and 2021, respectively, which represents a growth rate of 15.48% and 18.64% for 2015 and 2021, respectively. From the perspective of administrative villages, the overall level of land use for rural tourism land varied greatly. In 2021, the overall level of land use for rural tourism had a maximum value of 0.84 in Puxi Village, Puxi Township, and a minimum value of 0.12 in Zhuokun Village, Rongmuda Township, with a seven-fold difference between the two areas. In 2015, the difference between areas with the highest land use for rural tourism was 15.8 times greater than those with the lowest land use for rural tourism; the difference was 21.33 times in 2009.

**Table 1.** Descriptive statistics of the overall level of rural tourism land in the study area.

| Time | Dimensionality | Max. | Upper Quartile | Median | Lower Quartile | Min. | Standard Deviation | Coefficient of Variation | Average Value |
|------|----------------|------|----------------|--------|----------------|------|--------------------|--------------------------|---------------|
| 2021 | Facility Land | 0.90 | 0.30 | 0.43 | 0.61 | 0.12 | 0.22 | 0.47 | 0.47 |
| | Landscape land | 0.89 | 0.16 | 0.28 | 0.58 | 0.09 | 0.25 | 0.67 | 0.37 |
| | Comprehensive land | 0.84 | 0.27 | 0.40 | 0.51 | 0.12 | 0.19 | 0.45 | 0.42 |
| 2015 | Facility Land | 0.81 | 0.19 | 0.40 | 0.56 | 0.03 | 0.23 | 0.59 | 0.39 |
| | Landscape land | 0.78 | 0.13 | 0.24 | 0.49 | 0.04 | 0.22 | 0.70 | 0.32 |
| | Comprehensive land | 0.79 | 0.19 | 0.32 | 0.47 | 0.05 | 0.21 | 0.60 | 0.35 |
| 2009 | Facility Land | 0.60 | 0.18 | 0.30 | 0.43 | 0.00 | 0.16 | 0.53 | 0.31 |
| | Landscape land | 0.67 | 0.11 | 0.20 | 0.40 | 0.00 | 0.19 | 0.70 | 0.27 |
| | Comprehensive land | 0.64 | 0.16 | 0.25 | 0.41 | 0.03 | 0.16 | 0.55 | 0.29 |

### 3.1.2. Characteristics of Spatiotemporal Variations in Land Use for Rural Tourism Spatial Autocorrelation Analysis

To explain the spatiotemporal variations in land use for rural tourism in the study area, GeoDa software was used to calculate the global Moran's *I* index to determine whether there was spatial correlation in the land use for rural tourism. The results indicate that the level of land use for rural tourism in the study area had a significant positive spatial correlation (Figure 1). The global Moran's *I* values for the overall level of land use for rural tourism in 2009, 2015, and 2021 were 0.428, 0.383, and 0.256, respectively, reflecting a decreasing trend. The *Z* score of the global Moran's *I* in 2021 was 3.260, with a 99% confidence level; in 2015 and 2009, the scores were 4.556 and 5.061, respectively, which were both at a 1% level of significance. Among them, the global Moran's *I* values of land use for the rural tourism landscape were 0.162, 0.461, and 0.453, respectively, while those for rural tourism facilities were 0.289, 0.290, and 0.290, respectively. These values were all positive and significant, with a 99% confidence level.

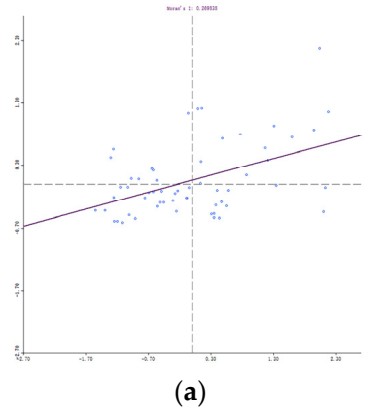 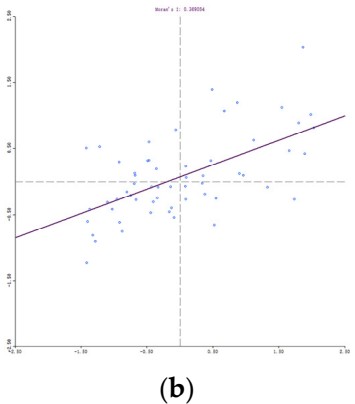 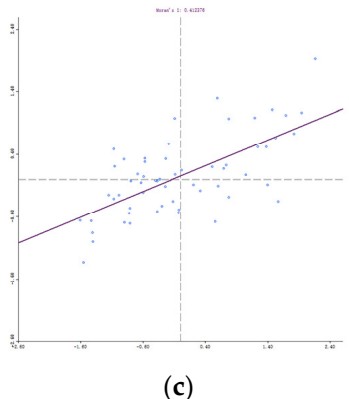

(a)           (b)           (c)

**Figure 1.** Moran scatter plot of the overall level of rural tourism land in the study area in 2021 (**a**), 2015 (**b**), 2009 (**c**).

Characteristics of Overall Level of Spatial Variations in Land Use for Rural Tourism

With the natural breaks method in the ArcGIS10.8 software, the level of land use for rural tourism was categorized into five levels: high, relatively high, moderate, relatively low, and low. The research results indicate that the overall level of spatial variations in land use for rural tourism in the study area from 2009 to 2021 was significant and distributed in clusters. These clusters were mainly centered on the county townships in the northwest and the river valley in the southeast, gradually expanding to the central region (Figure 2). Between 2009 and 2021, the number of administrative villages in the spatial clusters with high or relatively high overall levels of land use for tourism increased from 17 to 21. This was mainly driven by a radiation effect from the county's economic core, where land area for tourist facilities was much larger. The southeast region is located in the river valley in close proximity to the capital of Ngawa Tibetan and Qiang Autonomous Prefecture and Jinchuan County, a major tourist county. Administrative villages with low or relatively low levels of land use for rural tourism were mostly distributed in the northeastern and central-western regions, with small changes in their overall number. The growth of administrative villages with low levels of land use was slow, with obvious polarization. Administrative villages with moderate levels of land use for rural tourism were mainly distributed in the northwestern and southern-central regions.

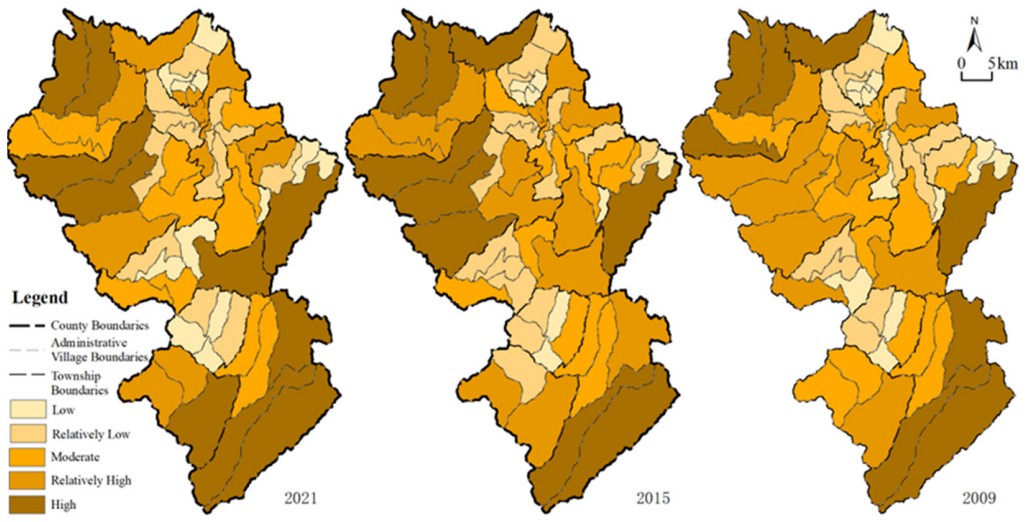

**Figure 2.** Spatial distribution of land use for rural tourism in 2021, 2015, and 2009.

Characteristics of Spatiotemporal Variations in Land Use for Rural Tourism Landscape

The research results indicate that the land use for the rural tourism landscape in the study area from 2009 to 2021 had a spatial distribution characterized by "two cores and four clusters". The northeast and southwest formed two core areas, with the four clusters gradually increasing towards the southeast (Figure 3). The spatial clusters in the two core areas were mainly characterized by low and relatively low levels of land use for the rural tourism landscape, including Yageche Village, Rongmuda Township, and Xingmuda in Gaduo Township in the northeastern region and Adou Village in Shili Township in the central-southern regions. The four spatial cluster areas were characterized by relatively high levels of land use for the rural tourism landscape, including Xiqiong Village in Shangduke Township in the northern region, Siyuewu Village in Puxi Township in the southern region, Dari Village in Gangmuda Township in the western region, and Kanglong Village in Zhongrangtang Township in the eastern region. The analysis shows that between 2009 and 2021, the growth rate of land use for the rural tourism landscape in the study area was relatively small, with a slightly higher number of villages with high levels of land use in 2021 than in 2009. However, the number of administrative villages with low levels of land use increased rapidly from 7 in 2009 to 15 in 2021.

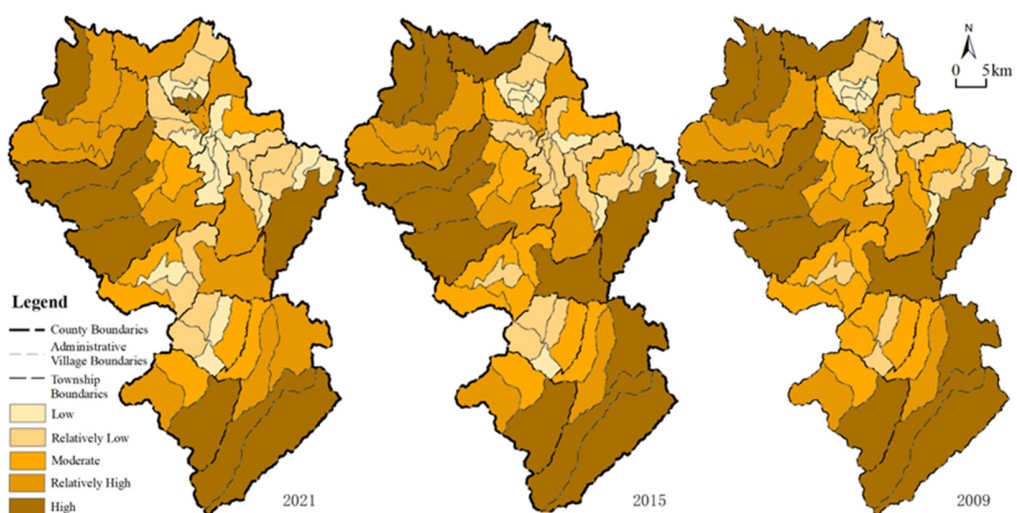

**Figure 3.** Spatial distribution of level of land use for rural tourism landscape in 2021, 2015, and 2009.

Characteristics of Spatiotemporal Variations in Land Use for Rural Tourism Facilities

From 2009 to 2021, the spatial distribution of land use for rural tourism facilities in the study area expanded to the central region from the two cores in the north and south, gradually forming a pattern of one cluster with multiple branches, with the cluster in the southwest and branches in the northwest, east, and south (Figure 4). Specifically, between 2009 and 2021, land use for rural tourism facilities in the southeast was high, including Siyuewu Village and Youri Village in Puxi Township. The number of administrative villages with high and relatively high levels of land use increased rapidly, gradually expanding to the northern region and formed a clustered development pattern. The branches with high levels of land use for rural tourism facilities were Yutuo Village in Shangduke Township in the northern region, Yangpei Village in Gangmuda Township in the western region, and Renpeng Village in Shangrangtang Township in the eastern region. These areas border Ganzi Prefecture, Sichuan Province, and Qinghai Province and are in close proximity to tourist attractions, such as the Lianbao Yeze and Seda Wuming Buddhist Academy. Areas with low and relatively low levels of land use were fairly concentrated and distributed in the northeastern and central regions, including Basheng Village and Yageche Village in Rongmuda Township.

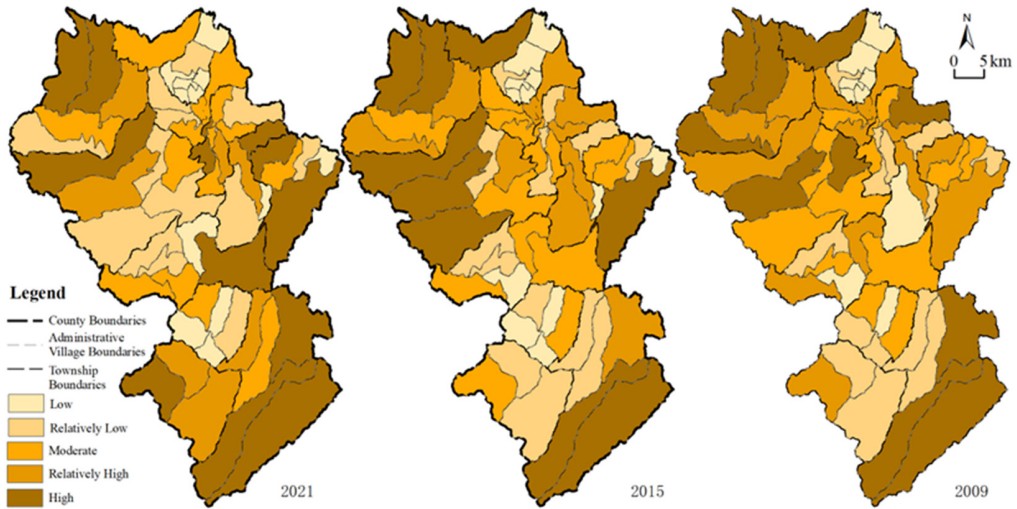

**Figure 4.** Spatial distribution of level of land use for rural tourism facilities in 2021, 2015, and 2009.

*3.2. Analysis of Driving Factors of Spatial Variations in Land Use for Rural Tourism*

3.2.1. Selection of Driving Factors

Land use for rural tourism is a branch of land use, with its spatial distribution mainly affected by natural conditions, the economic base, tourism resources, and locational factors [33]. Therefore, in this study, a five-in-one evaluation index system of natural conditions, society and economy, tourism resources, geographical location, and regional policies was constructed (Table 2).

**Table 2.** Evaluation index system of factors that influence the spatiotemporal characteristics of land use for rural tourism in the study area.

| Dimension | Indicators | Explanation and Source of Indicators |
|---|---|---|
| Natural Conditions | Average slope/ | Take the average value of slopes in the village with zoning statistics on ArcGIS |
| | Average altitude/km | ArcGIS raster statistics |
| Society and Economy | Per capita income of the village's collective economy/CNY 10,000 | Total income of the collective economy in the administrative village/number of permanent residents; sourced from the agricultural economic report |
| | Proportion of minority population/% | Minority population/permanent population |
| | Average years of education of village cadres/n | Sum of years of education of village cadres/total number of village cadres |
| Tourist Resources | Number of tourist attractions above a 3A rating | Number of scenic spots above 3A rating, sourced from survey data |
| | Number of farmhouses and homestays in the whole village/unit | Sum of farmhouses, homestays, hotels, restaurants, etc.; sourced from survey data |
| | Benefits driven by tourism/CNY 10,000 | Based on value assigned in survey: very large = 4, relatively large = 3, ordinary = 2, needs to be improved = 1 |
| Geographical Location | Time from the county town/h Time from Chengdu/h | Obtained based on Gaode map and social surveys Obtained based on Gaode map and social surveys |
| | Convenience of transportation to the outside/km | Sum of distances from administrative villages to expressways, provincial roads, and county roads obtained from ArcGIS software |
| Regional Policy | Whether it is a key village | Based on value assigned in the survey: poverty-stricken village/village that eliminated poverty/village under assistance of rural revitalization strategy = 2, otherwise = 1 |
| | Government investment/CNY 10,000 | Sum of capital investment from various high-level departments, sourced from the County Finance Bureau |

3.2.2. Analysis of Driving Factor Detection Results

Dimension of Natural Conditions

From 2009 to 2021, there were relatively large differences in the indicators grouped under the dimension of natural conditions, with the overall determining power exhibiting a downward trend (Table 3). Specifically, there was a significant correlation between the average slope and average elevation and the level of land use for rural tourism in 2009 at the 1% significance level, with the determining powers being 0.465 and 0.382, respectively. In 2015, the determining power of the average slope and average altitude on the level of land use for rural tourism was lower and did not pass the test. In 2021, the levels of land use for rural tourism and average slope were not significant, with a relatively small determining power. It is worth noting that although the relationship between average altitude and the level of land use for rural tourism was insignificant, it became significant in 2015 at the 10% level. This can be ascribed to better climatic conditions in areas with a higher altitude and more tourists in summer that benefited the development of tourism. Overall, the natural conditions in the study area from 2009 to 2021 had an impact on the level of land use for rural tourism, albeit with the determining power decreasing each year.

**Table 3.** Overview of detection and analysis of factors affecting land use for rural tourism in the study area.

| Dimension | Indicator | 2009 | | 2015 | | 2021 | |
|---|---|---|---|---|---|---|---|
| | | *q* Value | *p* Value | *q* Value | *p* Value | *q* Value | *p* Value |
| Natural Conditions | Average slope | 0.465 | 0.000 *** | 0.106 | 0.173 | 0.089 | 0.146 |
| | Average altitude | 0.382 | 0.000 *** | 0.111 | 0.134 | 0.138 | 0.092 * |
| Society and Economy | Per capita income of village's collective economy | 0.098 | 0.173 | 0.270 | 0.011 ** | 0.374 | 0.000 *** |
| | Proportion of minority population | 0.327 | 0.000 *** | 0.115 | 0.129 | 0.085 | 0.153 |
| | Average years of education of village cadres | 0.217 | 0.014 ** | 0.228 | 0.048 ** | 0.364 | 0.000 *** |
| Tourism Resources | Number of tourist attractions above a 3A rating | 0.233 | 0.024 ** | 0.321 | 0.000 *** | 0.481 | 0.000 *** |
| | Number of farmhouses and homestays in the whole village | 0.101 | 0.139 | 0.317 | 0.000 *** | 0.423 | 0.000 *** |
| | Tourism-driven benefits | 0.403 | 0.000 *** | 0.471 | 0.000 *** | 0.498 | 0.000 *** |
| Geographical Location | Time from the county town | 0.377 | 0.000 *** | 0.167 | 0.082 * | 0.176 | 0.062 * |
| | Time from Chengdu | 0.076 | 0.212 | 0.128 | 0.096 * | 0.103 | 0.107 |
| | Convenience of transportation to other regions | 0.144 | 0.099 * | 0.211 | 0.079 * | 0.320 | 0.000 *** |
| Regional Policy | Whether it is a key village | 0.106 | 0.122 | 0.207 | 0.061 * | 0.248 | 0.032 ** |
| | Government investment | 0.271 | 0.011 ** | 0.299 | 0.000 *** | 0.318 | 0.000 *** |

Note: ***, **, * indicate that the detected variable is significant at the level of 1%, 5%, or 10%, respectively, the *q* value indicates the determining power, and the *p* value indicates the significance.

Socioeconomic Dimension

From 2009 to 2021, the per capita income of the village's collective economy and the average years of education of village cadres gradually increased in determining power, while the proportion of minority population significantly weakened. Specifically, in 2009, there were significant correlations between the proportion of minority populations, the average years of education of village cadres, and the level of land use for rural tourism. However, the per capita income of the village's collective economy did not pass the significance test, which was mainly due to the lack of income from the collective economy in the county's 58 villages. In 2015, there were significant correlations between the per capita income of the village's collective economy and the level of land use for rural tourism, while the influence of the proportion of minority populations decreased and did not pass the significance test. The average years of education of village cadres was significant at the level of 5%, which was essentially consistent with 2009. Lastly, in 2021, the per capita income of the village's collective economy and the average years of education of village cadres gradually increased in their determining power, while the proportion of the minority populations was consistent with levels in 2015 and did not pass the significance test.

Dimension of Tourism Resources

From 2009 to 2021, the impact of tourism resources on the overall level of land use for rural tourism gradually increased. In 2009, there was a significant correlation between tourism-driven benefits, the number of tourist attractions with ratings of 3A and above, and the overall level of land use for rural tourism; however, the number of farmhouses and homestays in villages had little impact on the level of land use, and it did not pass the significance test. In 2015, there was a significant correlation between the level of land use for rural tourism and the three indicators of tourism-driven benefits, the number of tourist attractions with ratings of 3A and above, and the number of farmhouses and homestays in the whole village. All were significant at the 1% level, with their determining power significantly higher than for the other indicators. These three indicators were still significant at the 1% level in 2021, similar to 2015, and are determining factors of land use for rural

tourism. Therefore, it can be seen that from 2009 to 2021, tourism resources were the dominant factor in land use for rural tourism.

Dimension of Geographical Location

The indicators of geographical location varied greatly, with the time from the county town and convenience of transportation to the outside being the primary factors, while the time from Chengdu had a smaller determining power. Specifically, in 2009, the time from the county town and convenience of transportation significantly correlated with the level of land use for rural tourism at the level of 1% and 10%, respectively; however, the determining power of the time from Chengdu did not pass the significance test. In 2015, all three indicators were correlated with land use for rural tourism. That said, the time from the county town and the time from Chengdu were less important factors, while convenience of transportation had greater determining power. In 2021, convenience of transportation became the dominant factor, passing the significant test at the 1% level. The determining power of the time from the county town on the level of land use for rural tourism was 0.176, which was significant at the 10% level, but the time from Chengdu did not pass the significance test.

Dimension of Regional Policies

The impact of regional policies on land use for rural tourism gradually increased, especially in key villages. In 2009, whether villages were key villages had a small impact on land use for rural tourism and did not pass the significance test. Government investment had a greater impact on the level of land use for rural tourism, with a determining power of 0.271. In 2015, there was a correlation between whether villages were key villages, whether they received government investment, and land use for rural tourism. During this period, poverty-stricken villages had more opportunities to obtain various resources. Development was much slower in villages that were not experiencing poverty. In 2021, there was a significant correlation between whether villages were key villages, whether they received government investment, and the level of land use for rural tourism, with the determining power significant at the 5% and 10% levels, respectively. It is evident that regional policies had a greater impact on land use for rural tourism from 2009 to 2021, especially with government support for villages at all levels gradually becoming the focus of rural tourism development.

3.2.3. Dynamic Analysis of Driving Factors

This study used a model of the degree of dynamic change to explore the driving factors that evolved over time in two periods (2009–2015 and 2015–2021). The factors were divided into four categories: enhanced, weakened, stabilized, and hybrid types. Hybrid types were divided into the following three types: enhanced–weakened, weakened–stabilized and stabilized–enhanced (Table 4).

Enhanced factors mainly included per capita income in the village's collective economy, the number of tourist attractions with a rating above 3A, the number of farmhouses and homestays in the whole village, the convenience of regional transportation, and whether the village was a key village. Among these factors, the determining power of per capita income in a village's collective economy increased significantly and became the dominant factor driving land use for rural tourism. Furthermore, the number of tourist attractions with a rating above 3A increased, but the number of farmhouses and homestays in a village and whether the village was a key village had a declining determining power. Stabilized factors were mainly tourism-driven benefits and government investment. Although the determining power of tourism-driven benefits had declined, it remained a dominant factor driving land use for rural tourism.

**Table 4.** Classification of factors affecting land use for rural tourism in the study area.

| Detected Factors | 2009–2015 | | 2015–2021 | | Category Type |
|---|---|---|---|---|---|
| | Degree of Dynamic Change | Type of Change | Degree of Dynamic Change | Type of Change | |
| Average slope | −77.20% | Weakened Factor | −16.04% | Weakened Factor | Weakened |
| Average altitude | −66.49% | Weakened Factor | 7.81% | Stabilized Factor | Weakened–Stabilized |
| Per capita income of village's collective economy | 175.51% | Enhanced Factor | 38.52% | Enhanced Factor | Enhanced |
| Proportion of minority population | −64.83% | Weakened Factor | −26.09% | Weakened Factor | Weakened |
| Average years of education of village cadres | 5.07% | Stabilized Factor | 59.65% | Enhanced Factor | Stabilized–Enhanced |
| Number of tourist attractions above 3A rating | 37.77% | Enhanced Factor | 49.84% | Enhanced Factor | Enhanced |
| Number of farmhouses and homestays in the whole village | 213.86% | Enhanced Factor | 33.44% | Enhanced Factor | Enhanced |
| Tourism-driven benefits | 16.87% | Stabilized Factor | 5.73% | Stabilized Factor | Stabilized |
| Time from the county town | −55.70% | Weakened Factor | 5.39% | Stabilized Factor | Weakened–Stabilized |
| Time from Chengdu | 46.05% | Enhanced Factor | −7.21% | Weakened Factor | Enhanced–Weakened |
| Convenience of transportation to the outside | 46.53% | Enhanced Factor | 51.66% | Enhanced Factor | Enhanced |
| Whether it is a key village | 95.28% | Enhanced Factor | 19.81% | Enhanced Factor | Enhanced |
| Government investment | 10.33% | Stabilized Factor | 6.35% | Stabilized Factor | Stabilized |

Since 2009, the Chinese government has increased various forms of investment in administrative villages to alleviate poverty, stabilize poverty alleviation, and achieve rural revitalization, resulting in a greater impact of government investment on the level of land use for rural tourism. From the perspective of weakened factors, which included mainly the average slope and proportion of minority population, the impact of both factors on the level of land use for rural tourism had significantly weakened. The latter can be mainly ascribed to a general improvement in the overall quality of minority populations and the significant improvement in accepting new things. From the perspective of hybrid factors, the determining power of the time from Chengdu first increased and subsequently decreased, making it an enhanced–weakened type. Average altitude and distance from the county town were categorized as the weakened–stabilized type, with their determining power gradually increasing and tending toward a stable state. Village cadres' average years of education was classified as a stabilized–enhanced type, with the magnitude of change in determining power between 2009 and 2021 increasing sharply after a stable phase. This indicates that the management ability and holistic quality of village cadres gradually contributed to greater decision-making power in land use for rural tourism.

## 4. Discussion

The study area encompassed ethnic minority areas, alpine areas, and poverty-stricken areas—a typical area where ethnic minorities have been lifted out of poverty. The overall level of land use for rural tourism from 2009 to 2021 was relatively low. Combining the results of previous research, the governance path for similar areas should be formulated based on the driving factors. First, the governance pathway for enhanced types should focus on industrial development, supported by strategies to develop rural tourism with unique local characteristics and strengthened policy support. This will increase villages' per capita income from their collective economy and improve the level of land use for rural tourism. Second, the governance pathway for weakened types should focus on strengthening business training to improve the level of land use for rural tourism and the overall quality

of the people, thereby bolstering the rural revitalization strategy. Third, the governance pathway for stabilized factors should focus on innovating the "tourism + development" model and multiple financing channels, which will enhance the linkages of diversified benefits among the government, companies, and farmers. Fourth, the governance pathway for hybrid factors should focus on improving transportation facilities and the support system for cadres stationed in the village, thereby boosting the convenience of travel and the working ability of village committees.

Research on land use for rural tourism covers multiple disciplines, such as management, economics, geography, and sociology. Although it is a complex and comprehensive system, it remains limited by data availability and research methods, with several shortcomings. For instance, the research scale is relatively unidimensional, with a lack of multi-scalar research at the township, county, city, and provincial scale. The evaluation system of influencing factors has room for improvement, and the intention of farmers was not considered. In the future, the measurement methods of the level of land use for rural tourism will be further enriched, with the research scale expanded and the evaluation system of influencing factors further improved.

## 5. Conclusions

Based on long-term data at a village level, this study leveraged mathematical models to analyze the spatiotemporal characteristics of land use for rural tourism in areas that eliminated poverty and identified the driving factors of land use for rural change and their dynamic changes over time. The results provide a basis for a proposal for targeted land management measures for tourism to facilitate the development of multi-channel and comprehensive solutions to resolve bottlenecks in land use for rural tourism development. The study also promotes the high-quality development of rural tourism, broadens avenues for farmers to increase their income and gain affluence, and enhances intrinsic motivation for development in areas that eliminated poverty and people who have been lifted out of poverty.

This study argues that: (1) The overall level of land use for rural tourism in areas that eliminated poverty was low, with large regional differences. As counties and villages that eliminated poverty broke free from past labels and gained greater visibility, the overall level of land use for rural tourism in this region grew rapidly, with a 44.83% increase from 2009 to 2021. In addition, the growth rate of land use for rural tourism facilities was significantly higher than that for the tourism landscape. (2) There were significant spatial clusters of land use for rural tourism in areas that eliminated poverty, with an overall cluster-like distribution centered around the county towns in the northwest and the river valley in the southeast; these clusters gradually expanded to the central region. The spatial distribution of land use for the rural tourism landscape was characterized as "two cores and four clusters", with the two cores in the northeast and southwest, which gradually increased to four clusters in the southeast. Land use for rural tourism facilities gradually exhibited a distribution typified as "one cluster with multiple branches", where the southeastern region was the cluster, and the northwestern, eastern, and southern regions formed the branches. (3) The driving factors of the spatial differentiation in land use for rural tourism land in areas that eliminated poverty were diversified and dynamic. The dominant factors shifted from natural conditions and geographical location to the dimensions of socioeconomic factors, tourism resources, and regional policies. The dynamic changes in the driving factors were mainly classified as enhanced and hybrid types, with the per capita income in villages' collective economy, the average years of education of village cadres, the number of tourist attractions with ratings above 3A, the number of farmhouses and homestays in the whole village, tourism-driven benefits, the convenience of regional transportation, and government investment as the dominant factors.

**Author Contributions:** Conceptualization, Y.L. (Yuanli Liu) and Y.L. (Yan Liu); Validation, H.L.; Data curation, Y.L. (Yuanli Liu); Formal analysis, Y.L. (Yan Liu); Methodology, Y.L. (Yuanli Liu) and J.Q.; Supervision and project administration, H.L.; Writing—original draft, Y.L.; Writing—review and editing, Y.L. (Yuanli Liu) and H.L.; Visualization, Y.L. (Yan Liu). All authors have read and agreed to the published version of the manuscript.

**Funding:** This research was supported by the Natural Science Foundation of Chongqing (No. CSTB2022NSCQ-MSX0464), the research fund of Southwest University of Science and Technology (No. 19sx7106), and the National Social Science Foundation of China (No. 20BSH079).

**Institutional Review Board Statement:** Not applicable.

**Informed Consent Statement:** Not applicable.

**Data Availability Statement:** Not applicable.

**Acknowledgments:** Thank you to everyone who contributed to this study.

**Conflicts of Interest:** The authors declare no conflict of interest.

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
