# Peer review of "Characteristics of Spatiotemporal Variations and Driving Factors of Land Use for Rural Tourism in Areas That Eliminated Poverty"

_land, doi:10.3390/land12040910_

Round 1
Reviewer 1 Report
The article discusses the interesting topic of the influence of land use on rural tourism. However, the article has some shortcomings in the presentation of the results that need to be corrected:
Significant flaws are in the formatting - for example, the figures are in a different place in the text than their description. In some places, there are no separate paragraphs from the headings of other chapters, etc. These formatting errors need to be removed by the authors and checked by the editor.
Global Moran's I and Local Moran's I should be two separate formulas, each with its own number.
In Figures 1 - 3, the order of the categories in the legend should be from "low" to "high"; the "moderate" category should be in the middle.
In chapter 4.1.2.1, Spatial Autocorrelation Analysis, I would expect a map with spatial autocorrelation results. The presentation of the results in the form of the text seems insufficient to me.
More attention should be paid to references. Their formatting is not uniform and according to the model of the publishing house, and there are various typos in them.
Author Response
Dear Reviewer,
Thank you very much for your guidance on this article. We have made the following changes based on your suggestions and look forward to receiving your support. If there are still problems, we hope you will give us another chance and we will make the changes again.
Point 1: Revision notes on various formatting issues
Response 1: We have combed through and revised the formatting of the entire article to ensure that there are no problems, as described in the article revision.
Point 2: Revision of the “Global Moran's I and Local Moran's I should be two separate formulas” issue.
Response 2: Thank you very much for your opinion, each with its own number.
Point 3: Revision of the “In Figures 1 - 3, the order of the categories in the legend should be from ‘low’ to ‘high’ ” issue.
Response 3: We have replaced the diagram and checked the other legends, as described in the article revision.
Point 4: Revision of the “I would expect a map with spatial autocorrelation results” issue.
Response 4: We have added the “Figure 1.Moran scatter plot of the overall level of rural tourism land in the study area in 2021,2015,2009” section, as described in the article revision.
Point 5: Revision of the “Reference format” issue.
Response 5: Thank you very much for your opinion. We combed through the references to make sure there were no problems, as described in the article revision.
Finally, thank you again for your guidance on our paper and we will definitely continue to work hard, best wishes.

Reviewer 2 Report
The article suggests a current and attractive topic for the academy. The effort made is evident, but it requires profound adjustments, I hope you find the following observations helpful:
It would help if you used the keywords effectively by not using the same word (s) with the title so your manuscript will be more discoverable.
In line 104, how do you explain that your data’s reliability and validity tests of the survey questionnaire met the requirements of scientific research so the reader knows it is true? Please provide a brief explanation.
I'm afraid I have to disagree with your argument: “Furthermore, due to limitations on the length of this paper, the calculation process of the level of land use for rural tourism was not elaborated (lines 408-409).“ In my opinion, you should show the calculation process using all of your equations so the reader and other scientists can learn and understand where the numbers come from. For instance, you can use a table to resume your calculation.
I think you should put the Discussion chapter before the Conclusion chapter. Furthermore, you should give more broad content for your Discussion chapter. You can discuss every figure why the growth of land use looks to change inconsistently year by year (see Figure 1 below for example).
Please provide justification and references so readers don't think that the discussion you will provide is not just your assumption. You may also compare (or contrast) the study found with previous studies.
I think you should add more references about case studies that occur in other regions instead China so you can compare or contrast your finding.
I’m excited about the potential of your paper and think improving your work may help you define your contribution to the literature. I wish you all the best as you continue your work.

Author Response
Dear Reviewer,
Thank you very much for your guidance on this article. We have made the following changes based on your suggestions and look forward to receiving your support. If there are still problems, we hope you will give us another chance and we will make the changes again.
Point 1: Revision of the “if you used the keywords effectively by not using the same word (s) with the title so your manuscript will be more discoverable” issue.
Response 1: Thank you for your suggestion, we have replaced the keywords, the new keywords are land use for rural tourism, rural industry revitalization, spatial distribution characteristics and areas that eliminated poverty.
Point 2: Revision of the “how do you explain that your data’s reliability and validity tests of the survey questionnaire met the requirements of scientific research” issue.
Response 2: Thank you for your suggestion, We added the results of the reliability test and validity test, as described in the article revision.
Point 3: Revision of the “you should show the calculation process using all of your equations so the reader and other scientists can learn and understand where the numbers come from” issue.
Response 3: Thank you for your suggestion, we carried out the change, remove the deficiencies and add the display of the measurement process, as described in the article revision.
Point 4: Revision of the “I think you should put the Discussion chapter before the Conclusion chapter” issue.
Response 4: Thank you for your suggestion, We have revised the discussion section more substantially to further discuss the deeper implications of rural tourism land use change and to compare and analyze the results with our previous study. We believe that the discussion section is based on the conclusion, so we have not adjusted the position of “discussion” and “conclusion” for the time being, and if we really need to change it, we will adjust it further, as described in the article revision.
Point 5: Revision of the “I think you should add more references about case studies that occur in other regions instead China so you can compare or contrast your finding” issue.
Response 5: Thank you for your suggestion, we have updated our references and look forward to comparing them with cases from other countries. Also, studying rural tourism land in other countries is a future direction for the team.
Finally, thank you again for your guidance on our paper and we will definitely continue to work hard, best wishes.

Reviewer 3 Report
Tourism offers great opportunities to the regions where it develops. Properly developed, it creates new jobs, strengthens the local economy, and develops local infrastructure, and also helps to protect the natural environment and natural and cultural heritage, as well as can reduce and reduce poverty and inequality.
The increase in tourist traffic results in the development of accompanying tourist services. As a result, tourism competes for land with other areas of economic life, including agriculture. If the construction of tourist and recreational facilities is increased, the danger to these land resources and attractive landscapes increases.
The work is interesting, and worth publishing, but I have some questions that I did not get answers to while reading the article. Maybe the authors will consider them and, in addition to responding to the review, will also supplement the article with them.
The presented data show that agricultural land has increased, while forests and grassland and other areas have decreased, what was the reason for this?
Will the increase in the attractiveness and the development of tourism in this area not limit the access of farmers to the land that will be allocated for the development of tourist services?
What kind of tourist infrastructure will be developed in the studied areas? Isn't it worth using what is already there to limit the negative impact of tourist services on the local environment? The negative effects of tourism arise when the level of tourist use is greater than the capacity of the environment to cope with that use within acceptable limits of change, and uncontrolled tourism development is a threat to many areas, including rural areas.
How is land used for tourism purposes in the study area?
The authors did not show what the tourist traffic looks like in the study area. Is it small or is it already intense?
What is the tourist supply in the researched area, for example, how many agritourism farms are there?
This is a study based on the presented analysis, and in what direction should this region develop in order to reduce poverty on the one hand, but also to limit the negative impact of too extensive tourism development on all aspects of socio-economic life?
Author Response
Dear Reviewer,
Thank you very much for your guidance on this article. We have made the following changes based on your suggestions and look forward to receiving your support. If there are still problems, we hope you will give us another chance and we will make the changes again.
Point 1: Revision of the “The presented data show that agricultural land has increased, while forests and grassland and other areas have decreased” issue.
Response 1: Thank you for your question. The current situation has arisen because China has increased the protection of arable land and garden land and strictly controlled the area of land for construction. Also, we have a detailed explanation at the paper, as described in the article revision.
Point 2: Revision of the “Will the increase in the attractiveness and the development of tourism in this area not limit the access of farmers to the land that will be allocated for the development of tourist services?” issue.
Response 2: Thank you for your question, the results of the paper study show that the overall level of rural tourism land from 2009-2021 is low, but shows an increasing trend. It can be seen that the development of rural tourism will not affect the area of farmers rural tourism land, but will increase the area of their land.
Point 3: Revision of the “What kind of tourist infrastructure will be developed in the studied areas? Isn't it worth using what is already there to limit the negative impact of tourist services on the local environment?” issue.
Response 3: Thank you for your advice. At present, rural tourism in China is a way to realize the rural revitalization strategy. Rural tourism is still in the development stage and has relatively little impact on the environment, but Chinese government controls the sustainable development of rural tourism through village planning. This article mainly talks about the current situation and problems of land use, and fails to cover environmental factors, which is also the shortcoming of this article. In future research, we will focus on the impact of environment on rural tourism land use, thank you for your understanding. If we do need to include environmental factors, then we will further revise and improve it next.
Point 4: Revision of the “How is land used for tourism purposes in the study area?” issue.
Response 4: Thank you for your question. Based on the concept of land use for rural tourism, this study calculated the level of land use for rural tourism landscape using four indicators—fields, forest land, grassland, and water bodies. Besides the study calculated the level of land use for rural tourism facilities using five indicators—rural roads, farmhouses, homestays, village squares, and scenic spots. As described in the article revision.
Point 5: Revision of the “The authors did not show what the tourist traffic looks like in the study area. Is it small or is it already intense?” issue.
Response 5: Thanks to your suggestion, we have added the current traffic status of the study area to the “Study Area Overview”, as described in the article revision. Meanwhile, we used the indicator of “accessibility” to describe the traffic condition of the study area regarding to influence factors of rural tourism land use. As described in the article revision.
Point 6: Revision of the “What is the tourist supply in the researched area, for example, how many agritourism farms are there?” issue.
Response 6: Thank you for your question. When we analyzed the impact factors of rural tourism land, we added the “tourism resource dimension”, in which the “number of farmhouses and B&Bs in the village” indicator was used to illustrate tourism resources and tourism supply, and the original data could not be shown. If you think we need to show the original data, we will further revise it, thank you for your understanding. As described in the article revision.
Point 7: Revision of the “This is a study based on the presented analysis, and in what direction should this region develop in order to reduce poverty on the one hand, but also to limit the negative impact of too extensive tourism development on all aspects of socio-economic life?” issue.
Response 7: Thank you for your question. In the “Discussion” section of the paper, we have developed four development paths based on the influencing factors of rural tourism land use, but we have not been able to elaborate on them, which is the direction of our further study and research. If we need to elaborate on this part, we will definitely revise it properly, thank you.
Finally, thank you again for your guidance on our paper and we will definitely continue to work hard, best wishes.

Round 2
Reviewer 2 Report
Thank you for your revised article.